# Positive-Unlabeled Learning with Non-Negative Risk Estimator

**Ryuichi Kiryo**[1,2]   **Gang Niu**[1,2]   **Marthinus C. du Plessis**   **Masashi Sugiyama**[2,1]

[1]The University of Tokyo, 7-3-1 Hongo, Tokyo 113-0033, Japan
[2]RIKEN, 1-4-1 Nihonbashi, Tokyo 103-0027, Japan
{ kiryo@ms., gang@ms., sugi@ }k.u-tokyo.ac.jp

## Abstract

From only *positive* (P) and *unlabeled* (U) data, a binary classifier could be trained with PU learning, in which the state of the art is *unbiased PU learning*. However, if its model is very flexible, empirical risks on training data will go negative, and we will suffer from serious overfitting. In this paper, we propose a *non-negative risk estimator* for PU learning: when getting minimized, it is more robust against overfitting, and thus we are able to use very flexible models (such as deep neural networks) given limited P data. Moreover, we analyze the *bias*, *consistency*, and *mean-squared-error reduction* of the proposed risk estimator, and bound the *estimation error* of the resulting *empirical risk minimizer*. Experiments demonstrate that our risk estimator fixes the overfitting problem of its unbiased counterparts.

## 1  Introduction

*Positive-unlabeled* (PU) *learning* can be dated back to [1, 2, 3] and has been well studied since then. It mainly focuses on binary classification applied to retrieval and novelty or outlier detection tasks [4, 5, 6, 7], while it also has applications in matrix completion [8] and sequential data [9, 10].

Existing PU methods can be divided into two categories based on how U data is handled. The first category (e.g., [11, 12]) identifies possible *negative* (N) data in U data, and then performs ordinary supervised (PN) learning; the second (e.g., [13, 14]) regards U data as N data with smaller weights. The former heavily relies on the heuristics for identifying N data; the latter heavily relies on good choices of the weights of U data, which is computationally expensive to tune.

In order to avoid tuning the weights, *unbiased PU learning* comes into play as a subcategory of the second category. The milestone is [4], which regards a U data as weighted P and N data simultaneously. It might lead to *unbiased risk estimators*, if we unrealistically assume that the class-posterior probability is one for all P data.[1] A breakthrough in this direction is [15] for proposing the first unbiased risk estimator, and a more general estimator was suggested in [16] as a common foundation. The former is unbiased but non-convex for loss functions satisfying some symmetric condition; the latter is always unbiased, and it is further convex for loss functions meeting some linear-odd condition [17, 18]. PU learning based on these unbiased risk estimators is the current state of the art.

However, the unbiased risk estimators will give negative empirical risks, if the model being trained is very flexible. For the general estimator in [16], there exist three partial risks in the total risk (see Eq. (2) defined later), especially it has a negative risk regarding P data as N data to cancel the bias caused by regarding U data as N data. The worst case is that the model can realize any measurable function and the loss function is not upper bounded, so that the empirical risk is not lower bounded. This needs to be fixed since the original risk, which is the target to be estimated, is non-negative.

To this end, we propose a novel *non-negative risk estimator* that follows and improves on the state-of-the-art unbiased risk estimators mentioned above. This estimator can be used for two purposes: first, given some validation data (which are also PU data), we can use our estimator to evaluate the risk—for this case it is *biased* yet *optimal*, and for some symmetric losses, the *mean-squared-error reduction* is guaranteed; second, given some training data, we can use our estimator to train binary classifiers—for this case its risk minimizer possesses an *estimation error bound* of the same order as the risk minimizers corresponding to its unbiased counterparts [15, 16, 19].

In addition, we propose a *large-scale* PU learning algorithm for minimizing the unbiased and non-negative risk estimators. This algorithm accepts any *surrogate loss* and is based on *stochastic optimization*, e.g., [20]. Note that [21] is the only existing large-scale PU algorithm, but it only accepts a single surrogate loss from [16] and is based on *sequential minimal optimization* [22].

The rest of this paper is organized as follows. In Section 2 we review unbiased PU learning, and in Section 3 we propose non-negative PU learning. Theoretical analyses are carried out in Section 4, and experimental results are discussed in Section 5. Conclusions are given in Section 6.

## 2   Unbiased PU learning

In this section, we review unbiased PU learning [15, 16].

**Problem settings**   Let $X \in \mathbb{R}^d$ and $Y \in \{\pm 1\}$ ($d \in \mathbb{N}$) be the input and output random variables. Let $p(x, y)$ be the *underlying joint density* of $(X, Y)$, $p_\mathrm{p}(x) = p(x \mid Y = +1)$ and $p_\mathrm{n}(x) = p(x \mid Y = -1)$ be the *P and N marginals* (a.k.a. the P and N class-conditional densities), $p(x)$ be the *U marginal*, $\pi_\mathrm{p} = p(Y = +1)$ be the *class-prior probability*, and $\pi_\mathrm{n} = p(Y = -1) = 1 - \pi_\mathrm{p}$. $\pi_\mathrm{p}$ is assumed known throughout the paper; it can be estimated from P and U data [23, 24, 25, 26].

Consider the *two-sample problem setting* of PU learning [5]: two sets of data are sampled independently from $p_\mathrm{p}(x)$ and $p(x)$ as $\mathcal{X}_\mathrm{p} = \{x_i^\mathrm{p}\}_{i=1}^{n_\mathrm{p}} \sim p_\mathrm{p}(x)$ and $\mathcal{X}_\mathrm{u} = \{x_i^\mathrm{u}\}_{i=1}^{n_\mathrm{u}} \sim p(x)$, and a classifier needs to be trained from $\mathcal{X}_\mathrm{p}$ and $\mathcal{X}_\mathrm{u}$.[2] If it is PN learning as usual, $\mathcal{X}_\mathrm{n} = \{x_i^\mathrm{n}\}_{i=1}^{n_\mathrm{n}} \sim p_\mathrm{n}(x)$ rather than $\mathcal{X}_\mathrm{u}$ would be available and a classifier could be trained from $\mathcal{X}_\mathrm{p}$ and $\mathcal{X}_\mathrm{n}$.

**Risk estimators**   Unbiased PU learning relies on unbiased risk estimators. Let $g : \mathbb{R}^d \to \mathbb{R}$ be an arbitrary *decision function*, and $\ell : \mathbb{R} \times \{\pm 1\} \to \mathbb{R}$ be the *loss function*, such that the value $\ell(t, y)$ means the loss incurred by predicting an output $t$ when the ground truth is $y$. Denote by $R_\mathrm{p}^+(g) = \mathbb{E}_\mathrm{p}[\ell(g(X), +1)]$ and $R_\mathrm{n}^-(g) = \mathbb{E}_\mathrm{n}[\ell(g(X), -1)]$, where $\mathbb{E}_\mathrm{p}[\cdot] = \mathbb{E}_{X \sim p_\mathrm{p}}[\cdot]$ and $\mathbb{E}_\mathrm{n}[\cdot] = \mathbb{E}_{X \sim p_\mathrm{n}}[\cdot]$. Then, the *risk* of $g$ is $R(g) = \mathbb{E}_{(X,Y)\sim p(x,y)}[\ell(g(X), Y)] = \pi_\mathrm{p} R_\mathrm{p}^+(g) + \pi_\mathrm{n} R_\mathrm{n}^-(g)$. In PN learning, thanks to the availability of $\mathcal{X}_\mathrm{p}$ and $\mathcal{X}_\mathrm{n}$, $R(g)$ can be approximated directly by

$$\widehat{R}_\mathrm{pn}(g) = \pi_\mathrm{p} \widehat{R}_\mathrm{p}^+(g) + \pi_\mathrm{n} \widehat{R}_\mathrm{n}^-(g), \tag{1}$$

where $\widehat{R}_\mathrm{p}^+(g) = (1/n_\mathrm{p}) \sum_{i=1}^{n_\mathrm{p}} \ell(g(x_i^\mathrm{p}), +1)$ and $\widehat{R}_\mathrm{n}^-(g) = (1/n_\mathrm{n}) \sum_{i=1}^{n_\mathrm{n}} \ell(g(x_i^\mathrm{n}), -1)$. In PU learning, $\mathcal{X}_\mathrm{n}$ is unavailable, but $R_\mathrm{n}^-(g)$ can be approximated indirectly, as shown in [15, 16]. Denote by $R_\mathrm{p}^-(g) = \mathbb{E}_\mathrm{p}[\ell(g(X), -1)]$ and $R_\mathrm{u}^-(g) = \mathbb{E}_{X \sim p(x)}[\ell(g(X), -1)]$. As $\pi_\mathrm{n} p_\mathrm{n}(x) = p(x) - \pi_\mathrm{p} p_\mathrm{p}(x)$, we can obtain that $\pi_\mathrm{n} R_\mathrm{n}^-(g) = R_\mathrm{u}^-(g) - \pi_\mathrm{p} R_\mathrm{p}^-(g)$, and $R(g)$ can be approximated indirectly by

$$\widehat{R}_\mathrm{pu}(g) = \pi_\mathrm{p} \widehat{R}_\mathrm{p}^+(g) - \pi_\mathrm{p} \widehat{R}_\mathrm{p}^-(g) + \widehat{R}_\mathrm{u}^-(g), \tag{2}$$

where $\widehat{R}_\mathrm{p}^-(g) = (1/n_\mathrm{p}) \sum_{i=1}^{n_\mathrm{p}} \ell(g(x_i^\mathrm{p}), -1)$ and $\widehat{R}_\mathrm{u}^-(g) = (1/n_\mathrm{u}) \sum_{i=1}^{n_\mathrm{u}} \ell(g(x_i^\mathrm{u}), -1)$.

The *empirical risk estimators* in Eqs. (1) and (2) are *unbiased* and *consistent* w.r.t. all popular loss functions.[3] When they are used for evaluating the risk (e.g., in cross-validation), $\ell$ is by default the *zero-one loss*, namely $\ell_{01}(t, y) = (1 - \mathrm{sign}(ty))/2$; when used for training, $\ell_{01}$ is replaced with a *surrogate loss* [27]. In particular, [15] showed that if $\ell$ satisfies a *symmetric condition*:

$$\ell(t, +1) + \ell(t, -1) = 1, \tag{3}$$

we will have

$$\widehat{R}_{\mathrm{pu}}(g) = 2\pi_{\mathrm{p}}\widehat{R}_{\mathrm{p}}^{+}(g) + \widehat{R}_{\mathrm{u}}^{-}(g) - \pi_{\mathrm{p}}, \tag{4}$$

which can be minimized by separating $\mathcal{X}_{\mathrm{p}}$ and $\mathcal{X}_{\mathrm{u}}$ with ordinary cost-sensitive learning. An issue is $\widehat{R}_{\mathrm{pu}}(g)$ in (4) must be non-convex in $g$, since no $\ell(t,y)$ in (3) can be convex in $t$. [16] showed that $\widehat{R}_{\mathrm{pu}}(g)$ in (2) is convex in $g$, if $\ell(t,y)$ is convex in $t$ and meets a *linear-odd condition* [17, 18]:

$$\ell(t,+1) - \ell(t,-1) = -t. \tag{5}$$

Let $g$ be parameterized by $\theta$, then (5) leads to a convex optimization problem so long as $g$ is linear in $\theta$, for which the globally optimal solution can be obtained. Eq. (5) is not only sufficient but also necessary for the convexity, if $\ell$ is unary, i.e., $\ell(t,-1) = \ell(-t,+1)$.

**Justification**  Thanks to the unbiasedness, we can study *estimation error bounds* (EEB). Let $\mathcal{G}$ be the *function class*, and $\widehat{g}_{\mathrm{pn}}$ and $\widehat{g}_{\mathrm{pu}}$ be the *empirical risk minimizers* of $\widehat{R}_{\mathrm{pn}}(g)$ and $\widehat{R}_{\mathrm{pu}}(g)$. [19] proved EEB of $\widehat{g}_{\mathrm{pu}}$ is tighter than EEB of $\widehat{g}_{\mathrm{pn}}$ when $\pi_{\mathrm{p}}/\sqrt{n_{\mathrm{p}}} + 1/\sqrt{n_{\mathrm{u}}} < \pi_{\mathrm{n}}/\sqrt{n_{\mathrm{n}}}$, if (a) $\ell$ satisfies (3) and is *Lipschitz continuous*; (b) the *Rademacher complexity* of $\mathcal{G}$ decays in $\mathcal{O}(1/\sqrt{n})$ for data of size $n$ drawn from $p(x)$, $p_{\mathrm{p}}(x)$ or $p_{\mathrm{n}}(x)$.[4] In other words, under mild conditions, PU learning is likely to outperform PN learning when $\pi_{\mathrm{p}}/\sqrt{n_{\mathrm{p}}} + 1/\sqrt{n_{\mathrm{u}}} < \pi_{\mathrm{n}}/\sqrt{n_{\mathrm{n}}}$. This phenomenon has been observed in experiments [19] and is illustrated in Figure 1(a).

# 3  Non-negative PU learning

In this section, we propose the non-negative risk estimator and the large-scale PU algorithm.

## 3.1  Motivation

Let us look inside the aforementioned justification of unbiased PU (uPU) learning. Intuitively, the advantage comes from the transformation $\pi_{\mathrm{n}}R_{\mathrm{n}}^{-}(g) = R_{\mathrm{u}}^{-}(g) - \pi_{\mathrm{p}}R_{\mathrm{p}}^{-}(g)$. When we approximate $\pi_{\mathrm{n}}R_{\mathrm{n}}^{-}(g)$ from N data $\{x_i^{\mathrm{n}}\}_{i=1}^{n_{\mathrm{n}}}$, the convergence rate is $\mathcal{O}_p(\pi_{\mathrm{n}}/\sqrt{n_{\mathrm{n}}})$, where $\mathcal{O}_p$ denotes the order in probability; when we approximate $R_{\mathrm{u}}^{-}(g) - \pi_{\mathrm{p}}R_{\mathrm{p}}^{-}(g)$ from P data $\{x_i^{\mathrm{p}}\}_{i=1}^{n_{\mathrm{p}}}$ and U data $\{x_i^{\mathrm{u}}\}_{i=1}^{n_{\mathrm{u}}}$, the convergence rate becomes $\mathcal{O}_p(\pi_{\mathrm{p}}/\sqrt{n_{\mathrm{p}}} + 1/\sqrt{n_{\mathrm{u}}})$. As a result, we might benefit from a tighter *uniform deviation bound* when $\pi_{\mathrm{p}}/\sqrt{n_{\mathrm{p}}} + 1/\sqrt{n_{\mathrm{u}}} < \pi_{\mathrm{n}}/\sqrt{n_{\mathrm{n}}}$.

However, the critical assumption on the Rademacher complexity is indispensable, otherwise it will be difficult for EEB of $\widehat{g}_{\mathrm{pu}}$ to be tighter than EEB of $\widehat{g}_{\mathrm{pn}}$. If $\mathcal{G} = \{g \mid \|g\|_{\infty} \leq C_g\}$ where $C_g > 0$ is a constant, i.e., it has all measurable functions with some bounded norm, then $\mathfrak{R}_{n,q}(\mathcal{G}) = \mathcal{O}(1)$ for any $n$ and $q(x)$ and all bounds become trivial; moreover if $\ell$ is not bounded from above, $\widehat{R}_{\mathrm{pu}}(g)$ becomes not bounded from below, i.e., it may diverge to $-\infty$. Thus, in order to obtain high-quality $\widehat{g}_{\mathrm{pu}}$, $\mathcal{G}$ cannot be too complex, or equivalently the model of $g$ cannot be too flexible.

This argument is supported by an experiment as illustrated in Figure 1(b). A *multilayer perceptron* was trained for separating the even and odd digits of MNIST hand-written digits [29]. This model is so flexible that the number of parameters is 500 times more than the total number of P and N data. From Figure 1(b) we can see:

(A) on training data, the risks of uPU and PN both decrease, and uPU is faster than PN;
(B) on test data, the risk of PN decreases, whereas the risk of uPU does not; the risk of uPU is lower at the beginning but higher at the end than that of PN.

To sum up, the overfitting problem of uPU is serious, which evidences that in order to obtain high-quality $\widehat{g}_{\mathrm{pu}}$, the model of $g$ cannot be too flexible.

## 3.2  Non-negative risk estimator

Nevertheless, we have no choice sometimes: we are interested in using flexible models, while labeling more data is out of our control. Can we alleviate the overfitting problem with neither changing the model nor labeling more data?

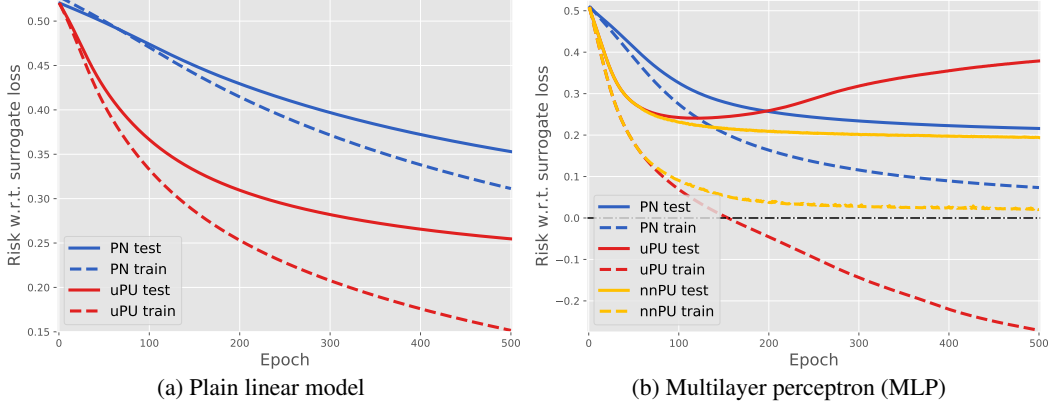

(a) Plain linear model                    (b) Multilayer perceptron (MLP)

The dataset is MNIST; even/odd digits are regarded as the P/N class, and $\pi_{\mathrm{p}} \approx 1/2$; $n_{\mathrm{p}} = 100$ and $n_{\mathrm{n}} = 50$ for PN learning; $n_{\mathrm{p}} = 100$ and $n_{\mathrm{u}} = 59,900$ for unbiased PU (uPU) and non-negative PU (nnPU) learning. The model is a plain linear model (784-1) in (a) and an MLP (784-100-1) with ReLU in (b); it was trained by Algorithm 1, where the loss $\ell$ is $\ell_{\mathrm{sig}}$, the optimization algorithm $\mathcal{A}$ is [20], with $\beta = 1/2$ for uPU, and $\beta = 0$ and $\gamma = 1$ for nnPU. Solid curves are $\widehat{R}_{\mathrm{pn}}(g)$ on test data where $g \in \{\widehat{g}_{\mathrm{pn}}, \widehat{g}_{\mathrm{pu}}, \widetilde{g}_{\mathrm{pu}}\}$, and dashed curves are $\widehat{R}_{\mathrm{pn}}(\widehat{g}_{\mathrm{pn}})$, $\widehat{R}_{\mathrm{pu}}(\widehat{g}_{\mathrm{pu}})$ and $\widetilde{R}_{\mathrm{pu}}(\widetilde{g}_{\mathrm{pu}})$ on training data. Note that nnPU is identical to uPU in (a).

Figure 1: Illustrative experimental results.

The answer is affirmative. Note that $\widehat{R}_{\mathrm{pu}}(\widehat{g}_{\mathrm{pu}})$ keeps decreasing and goes negative. This should be fixed since $R(g) \geq 0$ for any $g$. Specifically, it holds that $R_{\mathrm{u}}^-(g) - \pi_{\mathrm{p}} R_{\mathrm{p}}^-(g) = \pi_{\mathrm{n}} R_{\mathrm{n}}^-(g) \geq 0$, but $\widehat{R}_{\mathrm{u}}^-(g) - \pi_{\mathrm{p}} \widehat{R}_{\mathrm{p}}^-(g) \geq 0$ is not always true, which is a potential reason for uPU to overfit. Based on this key observation, we propose a *non-negative risk estimator* for PU learning:

$$\widetilde{R}_{\mathrm{pu}}(g) = \pi_{\mathrm{p}} \widehat{R}_{\mathrm{p}}^+(g) + \max\left\{0, \widehat{R}_{\mathrm{u}}^-(g) - \pi_{\mathrm{p}} \widehat{R}_{\mathrm{p}}^-(g)\right\}. \tag{6}$$

Let $\widetilde{g}_{\mathrm{pu}} = \arg\min_{g \in \mathcal{G}} \widetilde{R}_{\mathrm{pu}}(g)$ be the empirical risk minimizer of $\widetilde{R}_{\mathrm{pu}}(g)$. We refer to the process of obtaining $\widetilde{g}_{\mathrm{pu}}$ as *non-negative PU* (nnPU) *learning*. The implementation of nnPU will be given in Section 3.3, and theoretical analyses of $\widetilde{R}_{\mathrm{pu}}(g)$ and $\widetilde{g}_{\mathrm{pu}}$ will be given in Section 4.

Again, from Figure 1(b) we can see:

(A) on training data, the risk of nnPU first decreases and then becomes more and more flat, so that the risk of nnPU is closer to the risk of PN and farther from that of uPU; in short, the risk of nnPU does not go down with uPU after a certain epoch;

(B) on test data, the tendency is similar, but the risk of nnPU does not go up with uPU;

(C) at the end, nnPU achieves the lowest risk on test data.

In summary, nnPU works by explicitly constraining the training risk of uPU to be non-negative.

## 3.3   Implementation

A list of popular loss functions and their properties is shown in Table 1. Let $g$ be parameterized by $\theta$. If $g$ is linear in $\theta$, the losses satisfying (5) result in convex optimizations. However, if $g$ needs to be flexible, it will be highly nonlinear in $\theta$; then the losses satisfying (5) are not advantageous over others, since the optimizations are anyway non-convex. In [15], the *ramp loss* was used and $\widehat{R}_{\mathrm{pu}}(g)$ was minimized by *the concave-convex procedure* [30]. This solver is fairly sophisticated, and if we replace $\widehat{R}_{\mathrm{pu}}(g)$ with $\widetilde{R}_{\mathrm{pu}}(g)$, it will be more difficult to implement. To this end, we propose to use the *sigmoid loss* $\ell_{\mathrm{sig}}(t, y) = 1/(1 + \exp(ty))$: its gradient is everywhere non-zero and $\widetilde{R}_{\mathrm{pu}}(g)$ can be minimized by off-the-shelf gradient methods.

In front of big data, we should scale PU learning up by stochastic optimization. Minimizing $\widehat{R}_{\mathrm{pu}}(g)$ is *embarrassingly parallel* while minimizing $\widetilde{R}_{\mathrm{pu}}(g)$ is not, since $\widehat{R}_{\mathrm{pu}}(g)$ is *point-wise* but $\widetilde{R}_{\mathrm{pu}}(g)$ is not due to the max operator. That being said, $\max\{0, \widehat{R}_{\mathrm{u}}^-(g; \mathcal{X}_{\mathrm{u}}) - \pi_{\mathrm{p}} \widehat{R}_{\mathrm{p}}^-(g; \mathcal{X}_{\mathrm{p}})\}$ is no greater than $(1/N) \sum_{i=1}^N \max\{0, \widehat{R}_{\mathrm{u}}^-(g; \mathcal{X}_{\mathrm{u}}^i) - \pi_{\mathrm{p}} \widehat{R}_{\mathrm{p}}^-(g; \mathcal{X}_{\mathrm{p}}^i)\}$, where $(\mathcal{X}_{\mathrm{p}}^i, \mathcal{X}_{\mathrm{u}}^i)$ is the $i$-th mini-batch, and hence the corresponding upper bound of $\widetilde{R}_{\mathrm{pu}}(g)$ can easily be minimized in parallel.

Table 1: Loss functions for PU learning and their properties.

| Name | Definition | (3) | (5) | Bounded | Lipschitz | $\ell'(z) \neq 0$ |
|------|-----------|-----|-----|---------|-----------|-------------------|
| Zero-one loss | $(1 - \mathrm{sign}(z))/2$ | ✓ | × | ✓ | × | $z = 0$ |
| Ramp loss | $\max\{0, \min\{1, (1-z)/2\}\}$ | ✓ | × | ✓ | ✓ | $z \in [-1, +1]$ |
| Squared loss | $(z-1)^2/4$ | × | ✓ | × | × | $z \in \mathbb{R}$ |
| Logistic loss | $\ln(1 + \exp(-z))$ | × | ✓ | × | ✓ | $z \in \mathbb{R}$ |
| Hinge loss | $\max\{0, 1-z\}$ | × | × | × | ✓ | $z \in (-\infty, +1]$ |
| Double hinge loss | $\max\{0, (1-z)/2, -z\}$ | × | ✓ | × | ✓ | $z \in (-\infty, +1]$ |
| Sigmoid loss | $1/(1 + \exp(z))$ | ✓ | × | ✓ | ✓ | $z \in \mathbb{R}$ |

All loss functions are unary, such that $\ell(t, y) = \ell(z)$ with $z = ty$. The ramp loss comes from [15]; the double hinge loss is from [16], in which the squared, logistic and hinge losses were discussed as well. The ramp and squared losses are scaled to satisfy (3) or (5). The sigmoid loss is a horizontally mirrored *logistic function*; the logistic loss is the negative logarithm of the logistic function.

---

**Algorithm 1** Large-scale PU learning based on stochastic optimization

---

    **Input:** training data $(\mathcal{X}_\mathrm{p}, \mathcal{X}_\mathrm{u})$;
           hyperparameters $0 \leq \beta \leq \pi_\mathrm{p} \sup_t \max_y \ell(t, y)$ and $0 \leq \gamma \leq 1$
    **Output:** model parameter $\theta$ for $\widehat{g}_\mathrm{pu}(x; \theta)$ or $\widetilde{g}_\mathrm{pu}(x; \theta)$
1:   Let $\mathcal{A}$ be an external SGD-like stochastic optimization algorithm such as [20] or [31]
2:   **while** no stopping criterion has been met:
3:      Shuffle $(\mathcal{X}_\mathrm{p}, \mathcal{X}_\mathrm{u})$ into $N$ mini-batches, and denote by $(\mathcal{X}_\mathrm{p}^i, \mathcal{X}_\mathrm{u}^i)$ the $i$-th mini-batch
4:      **for** $i = 1$ **to** $N$:
5:          **if** $\widehat{R}_\mathrm{u}^-(g; \mathcal{X}_\mathrm{u}^i) - \pi_\mathrm{p}\widehat{R}_\mathrm{p}^-(g; \mathcal{X}_\mathrm{p}^i) \geq -\beta$:
6:             Set gradient $\nabla_\theta \widehat{R}_\mathrm{pu}(g; \mathcal{X}_\mathrm{p}^i, \mathcal{X}_\mathrm{u}^i)$
7:             Update $\theta$ by $\mathcal{A}$ with its current step size $\eta$
8:          **else**:
9:             Set gradient $\nabla_\theta(\pi_\mathrm{p}\widehat{R}_\mathrm{p}^-(g; \mathcal{X}_\mathrm{p}^i) - \widehat{R}_\mathrm{u}^-(g; \mathcal{X}_\mathrm{u}^i))$
10:         Update $\theta$ by $\mathcal{A}$ with a discounted step size $\gamma\eta$

---

The large-scale PU algorithm is described in Algorithm 1. Let $r_i = \widehat{R}_\mathrm{u}^-(g; \mathcal{X}_\mathrm{u}^i) - \pi_\mathrm{p}\widehat{R}_\mathrm{p}^-(g; \mathcal{X}_\mathrm{p}^i)$. In practice, we may tolerate $r_i \geq -\beta$ where $0 \leq \beta \leq \pi_\mathrm{p}\sup_t \max_y \ell(t, y)$, as $r_i$ comes from a single mini-batch. The degree of tolerance is controlled by $\beta$: there is zero tolerance if $\beta = 0$, and we are minimizing $\widehat{R}_\mathrm{pu}(g)$ if $\beta = \pi_\mathrm{p}\sup_t \max_y \ell(t, y)$. Otherwise if $r_i < -\beta$, we go along $-\nabla_\theta r_i$ with a step size discounted by $\gamma$ where $0 \leq \gamma \leq 1$, to make this mini-batch less overfitted. Algorithm 1 is insensitive to the choice of $\gamma$, if the optimization algorithm $\mathcal{A}$ is adaptive such as [20] or [31].

## 4 Theoretical analyses

In this section, we analyze the risk estimator (6) and its minimizer (all proofs are in Appendix B).

### 4.1 Bias and consistency

Fix $g$, $\widetilde{R}_\mathrm{pu}(g) \geq \widehat{R}_\mathrm{pu}(g)$ for any $(\mathcal{X}_\mathrm{p}, \mathcal{X}_\mathrm{u})$ but $\widehat{R}_\mathrm{pu}(g)$ is unbiased, which implies $\widetilde{R}_\mathrm{pu}(g)$ is biased in general. A fundamental question is then whether $\widetilde{R}_\mathrm{pu}(g)$ is consistent. From now on, we prove this consistency. To begin with, partition all possible $(\mathcal{X}_\mathrm{p}, \mathcal{X}_\mathrm{u})$ into $\mathfrak{D}^+(g) = \{(\mathcal{X}_\mathrm{p}, \mathcal{X}_\mathrm{u}) \mid \widehat{R}_\mathrm{u}^-(g) - \pi_\mathrm{p}\widehat{R}_\mathrm{p}^-(g) \geq 0\}$ and $\mathfrak{D}^-(g) = \{(\mathcal{X}_\mathrm{p}, \mathcal{X}_\mathrm{u}) \mid \widehat{R}_\mathrm{u}^-(g) - \pi_\mathrm{p}\widehat{R}_\mathrm{p}^-(g) < 0\}$. Assume there are $C_g > 0$ and $C_\ell > 0$ such that $\sup_{g \in \mathcal{G}} \|g\|_\infty \leq C_g$ and $\sup_{|t| \leq C_g} \max_y \ell(t, y) \leq C_\ell$.

**Lemma 1.** *The following three conditions are equivalent: (A) the probability measure of $\mathfrak{D}^-(g)$ is non-zero; (B) $\widetilde{R}_\mathrm{pu}(g)$ differs from $\widehat{R}_\mathrm{pu}(g)$ with a non-zero probability over repeated sampling of $(\mathcal{X}_\mathrm{p}, \mathcal{X}_\mathrm{u})$; (C) the bias of $\widetilde{R}_\mathrm{pu}(g)$ is positive. In addition, by assuming that there is $\alpha > 0$ such that $R_\mathrm{n}^-(g) \geq \alpha$, the probability measure of $\mathfrak{D}^-(g)$ can be bounded by*

$$\Pr(\mathfrak{D}^-(g)) \leq \exp(-2(\alpha/C_\ell)^2/(\pi_\mathrm{p}^2/n_\mathrm{p} + 1/n_\mathrm{u})). \tag{7}$$

Based on Lemma 1, we can show the exponential decay of the bias and also the consistency. For convenience, denote by $\chi_{n_\mathrm{p},n_\mathrm{u}} = 2\pi_\mathrm{p}/\sqrt{n_\mathrm{p}} + 1/\sqrt{n_\mathrm{u}}$.

**Theorem 2** (Bias and consistency). *Assume that $R_\mathrm{n}^-(g) \geq \alpha > 0$ and denote by $\Delta_g$ the right-hand side of Eq.* (7). *As $n_\mathrm{p}, n_\mathrm{u} \to \infty$, the bias of $\widetilde{R}_\mathrm{pu}(g)$ decays exponentially:*

$$0 \leq \mathbb{E}_{\mathcal{X}_\mathrm{p}, \mathcal{X}_\mathrm{u}}[\widetilde{R}_\mathrm{pu}(g)] - R(g) \leq C_\ell \pi_\mathrm{p} \Delta_g. \tag{8}$$

*Moreover, for any $\delta > 0$, let $C_\delta = C_\ell \sqrt{\ln(2/\delta)/2}$, then we have with probability at least $1 - \delta$,*

$$|\widetilde{R}_\mathrm{pu}(g) - R(g)| \leq C_\delta \cdot \chi_{n_\mathrm{p},n_\mathrm{u}} + C_\ell \pi_\mathrm{p} \Delta_g, \tag{9}$$

*and with probability at least $1 - \delta - \Delta_g$,*

$$|\widetilde{R}_\mathrm{pu}(g) - R(g)| \leq C_\delta \cdot \chi_{n_\mathrm{p},n_\mathrm{u}}. \tag{10}$$

Either (9) or (10) in Theorem 2 indicates for fixed $g$, $\widetilde{R}_\mathrm{pu}(g) \to R(g)$ in $\mathcal{O}_p(\pi_\mathrm{p}/\sqrt{n_\mathrm{p}} + 1/\sqrt{n_\mathrm{u}})$. This convergence rate is optimal according to the *central limit theorem* [32], which means the proposed estimator is a biased yet optimal estimator to the risk.

## 4.2 Mean squared error

After introducing the bias, $\widetilde{R}_\mathrm{pu}(g)$ tends to overestimate $R(g)$. It is not a *shrinkage estimator* [33, 34] so that its *mean squared error* (MSE) is not necessarily smaller than that of $\widehat{R}_\mathrm{pu}(g)$. However, we can still characterize this reduction in MSE.

**Theorem 3** (MSE reduction). *It holds that* $\mathrm{MSE}(\widetilde{R}_\mathrm{pu}(g)) < \mathrm{MSE}(\widehat{R}_\mathrm{pu}(g))$,[5] *if and only if*

$$\int_{(\mathcal{X}_\mathrm{p}, \mathcal{X}_\mathrm{u}) \in \mathfrak{D}^-(g)} (\widehat{R}_\mathrm{pu}(g) + \widetilde{R}_\mathrm{pu}(g) - 2R(g))(\widehat{R}_\mathrm{u}^-(g) - \pi_\mathrm{p}\widehat{R}_\mathrm{p}^-(g)) \, \mathrm{d}F(\mathcal{X}_\mathrm{p}, \mathcal{X}_\mathrm{u}) > 0, \tag{11}$$

*where $\mathrm{d}F(\mathcal{X}_\mathrm{p}, \mathcal{X}_\mathrm{u}) = \prod_{i=1}^{n_\mathrm{p}} p_\mathrm{p}(x_i^\mathrm{p})\mathrm{d}x_i^\mathrm{p} \cdot \prod_{i=1}^{n_\mathrm{u}} p(x_i^\mathrm{u})\mathrm{d}x_i^\mathrm{u}$. Eq. (11) is valid, if the following conditions are met: (a) $\Pr(\mathfrak{D}^-(g)) > 0$; (b) $\ell$ satisfies Eq.* (3); *(c) $R_\mathrm{n}^-(g) \geq \alpha > 0$; (d) $n_\mathrm{u} \gg n_\mathrm{p}$, such that we have $R_\mathrm{u}^-(g) - \widehat{R}_\mathrm{u}^-(g) \leq 2\alpha$ almost surely on $\mathfrak{D}^-(g)$. In fact, given these four conditions, we have for any $0 \leq \beta \leq C_\ell \pi_\mathrm{p}$,*

$$\mathrm{MSE}(\widehat{R}_\mathrm{pu}(g)) - \mathrm{MSE}(\widetilde{R}_\mathrm{pu}(g)) \geq 3\beta^2 \Pr\{\widetilde{R}_\mathrm{pu}(g) - \widehat{R}_\mathrm{pu}(g) > \beta\}. \tag{12}$$

The assumption (d) in Theorem 3 is explained as follows. Since *U data can be much cheaper than P data* in practice, it would be natural to assume $n_\mathrm{u}$ is much larger and grows much faster than $n_\mathrm{p}$, hence $\Pr\{R_\mathrm{u}^-(g) - \widehat{R}_\mathrm{u}^-(g) \geq \alpha\}/\Pr\{\widehat{R}_\mathrm{p}^-(g) - R_\mathrm{p}^-(g) \geq \alpha/\pi_\mathrm{p}\} \propto \exp(n_\mathrm{p} - n_\mathrm{u})$ asymptotically.[6] This means the contribution of $\mathcal{X}_\mathrm{u}$ is negligible for making $(\mathcal{X}_\mathrm{p}, \mathcal{X}_\mathrm{u}) \in \mathfrak{D}^-(g)$ so that $\Pr(\mathfrak{D}^-(g))$ exhibits exponential decay mainly in $n_\mathrm{p}$. As $\Pr\{R_\mathrm{u}^-(g) - \widehat{R}_\mathrm{u}^-(g) \geq 2\alpha\}$ has stronger exponential decay in $n_\mathrm{u}$ than $\Pr\{R_\mathrm{u}^-(g) - \widehat{R}_\mathrm{u}^-(g) \geq \alpha\}$ as well as $n_\mathrm{u} \gg n_\mathrm{p}$, we made the assumption (d).

## 4.3 Estimation error

While Theorems 2 and 3 addressed the use of (6) for evaluating the risk, we are likewise interested in its use for training classifiers. In what follows, we analyze the estimation error $R(\widetilde{g}_\mathrm{pu}) - R(g^*)$, where $g^*$ is the true risk minimizer in $\mathcal{G}$, i.e., $g^* = \arg\min_{g \in \mathcal{G}} R(g)$. As a common practice [28], assume that $\ell(t, y)$ is Lipschitz continuous in $t$ for all $|t| \leq C_g$ with a Lipschitz constant $L_\ell$.

**Theorem 4** (Estimation error bound). *Assume that (a) $\inf_{g \in \mathcal{G}} R_\mathrm{n}^-(g) \geq \alpha > 0$ and denote by $\Delta$ the right-hand side of Eq.* (7); *(b) $\mathcal{G}$ is closed under negation, i.e., $g \in \mathcal{G}$ if and only if $-g \in \mathcal{G}$. Then, for any $\delta > 0$, with probability at least $1 - \delta$,*

$$R(\widetilde{g}_\mathrm{pu}) - R(g^*) \leq 16L_\ell \pi_\mathrm{p} \mathfrak{R}_{n_\mathrm{p},p_\mathrm{p}}(\mathcal{G}) + 8L_\ell \mathfrak{R}_{n_\mathrm{u},p}(\mathcal{G}) + 2C_\delta' \cdot \chi_{n_\mathrm{p},n_\mathrm{u}} + 2C_\ell \pi_\mathrm{p} \Delta, \tag{13}$$

*where $C_\delta' = C_\ell \sqrt{\ln(1/\delta)/2}$, and $\mathfrak{R}_{n_\mathrm{p},p_\mathrm{p}}(\mathcal{G})$ and $\mathfrak{R}_{n_\mathrm{u},p}(\mathcal{G})$ are the Rademacher complexities of $\mathcal{G}$ for the sampling of size $n_\mathrm{p}$ from $p_\mathrm{p}(x)$ and of size $n_\mathrm{u}$ from $p(x)$, respectively.*

Theorem 4 ensures that learning with (6) is also consistent: as $n_\mathrm{p}, n_\mathrm{u} \to \infty$, $R(\widetilde{g}_\mathrm{pu}) \to R(g^*)$ and if $\ell$ satisfies (5), all optimizations are convex and $\widetilde{g}_\mathrm{pu} \to g^*$. For linear-in-parameter models with a bounded norm, $\mathfrak{R}_{n_\mathrm{p},p_\mathrm{p}}(\mathcal{G}) = \mathcal{O}(1/\sqrt{n_\mathrm{p}})$ and $\mathfrak{R}_{n_\mathrm{u},p}(\mathcal{G}) = \mathcal{O}(1/\sqrt{n_\mathrm{u}})$, and thus $R(\widetilde{g}_\mathrm{pu}) \to R(g^*)$ in $\mathcal{O}_p(\pi_\mathrm{p}/\sqrt{n_\mathrm{p}} + 1/\sqrt{n_\mathrm{u}})$.

For comparison, $R(\widehat{g}_\mathrm{pu}) - R(g^*)$ can be bounded using a *different proof technique* [19]:

$$R(\widehat{g}_\mathrm{pu}) - R(g^*) \le 8L_\ell \pi_\mathrm{p} \mathfrak{R}_{n_\mathrm{p},p_\mathrm{p}}(\mathcal{G}) + 4L_\ell \mathfrak{R}_{n_\mathrm{u},p}(\mathcal{G}) + 2C_\delta \cdot \chi_{n_\mathrm{p},n_\mathrm{u}}, \tag{14}$$

where $C_\delta = C_\ell\sqrt{\ln(2/\delta)/2}$. The differences of (13) and (14) are completely from the differences of the corresponding uniform deviation bounds, i.e., the following lemma and Lemma 8 of [19].

**Lemma 5.** *Under the assumptions of Theorem 4, for any $\delta > 0$, with probability at least $1 - \delta$,*

$$\sup_{g \in \mathcal{G}} |\widetilde{R}_\mathrm{pu}(g) - R(g)| \le 8L_\ell \pi_\mathrm{p} \mathfrak{R}_{n_\mathrm{p},p_\mathrm{p}}(\mathcal{G}) + 4L_\ell \mathfrak{R}_{n_\mathrm{u},p}(\mathcal{G}) + C_\delta' \cdot \chi_{n_\mathrm{p},n_\mathrm{u}} + C_\ell \pi_\mathrm{p} \Delta. \tag{15}$$

Notice that $\widehat{R}_\mathrm{pu}(g)$ is point-wise while $\widetilde{R}_\mathrm{pu}(g)$ is not due to the maximum, which makes Lemma 5 much more difficult to prove than Lemma 8 of [19]. The key trick is that after *symmetrization*, we employ $|\max\{0, z\} - \max\{0, z'\}| \le |z - z'|$, making three differences of partial risks point-wise (see (18) in the proof). As a consequence, we have to use a different Rademacher complexity *with the absolute value inside the supremum* [35, 36], whose *contraction* makes the coefficients of (15) doubled compared with Lemma 8 of [19]; moreover, we have to assume $\mathcal{G}$ is closed under negation to change back to the standard Rademacher complexity *without the absolute value* [28]. Therefore, the differences of (13) and (14) are mainly due to different proof techniques and cannot reflect the intrinsic differences of empirical risk minimizers.

# 5 Experiments

In this section, we compare PN, unbiased PU (uPU) and non-negative PU (nnPU) learning experimentally. We focus on training deep neural networks, as uPU learning usually does not overfit if a linear-in-parameter model is used [19] and nothing needs to be fixed.

Table 2 describes the specification of benchmark datasets. MNIST, 20News and CIFAR-10 have 10, 7 and 10 classes originally, and we constructed the P and N classes from them as follows: MNIST was preprocessed in such a way that 0, 2, 4, 6, 8 constitute the P class, while 1, 3, 5, 7, 9 constitute the N class; for 20News, 'alt.', 'comp.', 'misc.' and 'rec.' make up the P class, and 'sci.', 'soc.' and 'talk.' make up the N class; for CIFAR-10, the P class is formed by 'airplane', 'automobile', 'ship' and 'truck', and the N class is formed by 'bird', 'cat', 'deer', 'dog', 'frog' and 'horse'. The dataset epsilon has 2 classes and such a construction is unnecessary.

Three learning methods were set up as follows: (A) for PN, $n_p = 1,000$ and $n_n = (\pi_\mathrm{n}/2\pi_\mathrm{p})^2 n_p$; (B) for uPU, $n_p = 1,000$ and $n_u$ is the total number of training data; (C) for nnPU, $n_p$ and $n_u$ are exactly same as uPU. For uPU and nnPU, P and U data were dependent, because neither $\widehat{R}_\mathrm{pu}(g)$ in Eq. (2) nor $\widetilde{R}_\mathrm{pu}(g)$ in Eq. (6) requires them to be independent. The choice of $n_n$ was motivated by [19] and may make nnPU potentially better than PN as $n_\mathrm{u} \to \infty$ (whether $n_\mathrm{p} < \infty$ or $n_\mathrm{p} \le n_\mathrm{u}$).

The model for MNIST was a 6-layer *multilayer perceptron* (MLP) with ReLU [40] (more specifically, $d$-300-300-300-300-1). For epsilon, the model was similar while the activation was replaced with Softsign [41] for better performance. For 20News, we borrowed the pre-trained word embeddings from GloVe [42], and the model can be written as $d$-avg_pool(word_emb($d$,300))-300-300-1,

Table 2: Specification of benchmark datasets, models, and optimition algorithms.

| Name | # Train | # Test | # Feature | $\pi_\mathrm{p}$ | Model $g(x;\theta)$ | Opt. alg. $\mathcal{A}$ |
|---|---|---|---|---|---|---|
| MNIST [29] | 60,000 | 10,000 | 784 | 0.49 | 6-layer MLP with ReLU | Adam [20] |
| epsilon [37] | 400,000 | 100,000 | 2,000 | 0.50 | 6-layer MLP with Softsign | Adam [20] |
| 20News [38] | 11,314 | 7,532 | 61,188 | 0.44 | 5-layer MLP with Softsign | AdaGrad [31] |
| CIFAR-10 [39] | 50,000 | 10,000 | 3,072 | 0.40 | 13-layer CNN with ReLU | Adam [20] |

See http://yann.lecun.com/exdb/mnist/ for MNIST, https://www.csie.ntu.edu.tw/~cjlin/libsvmtools/datasets/binary.html for epsilon, http://qwone.com/~jason/20Newsgroups/ for 20Newsgroups, and https://www.cs.toronto.edu/~kriz/cifar.html for CIFAR-10.

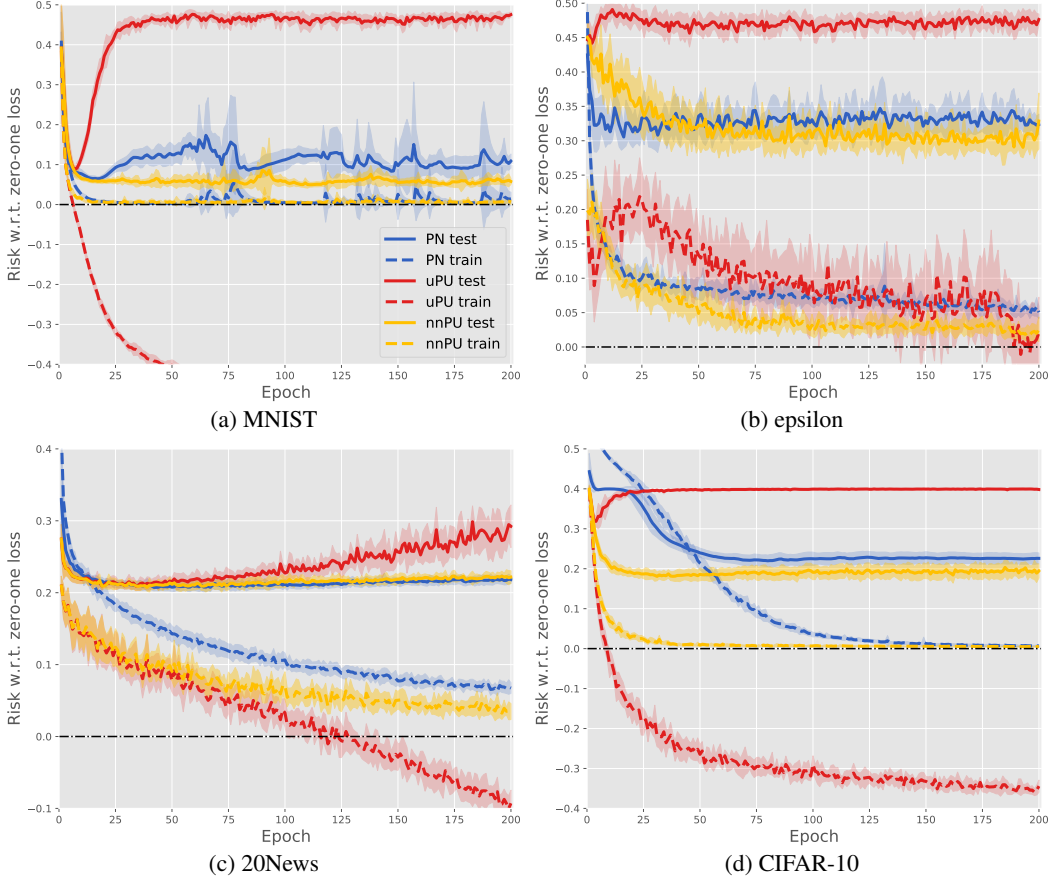

Figure 2: Experimental results of training deep neural networks.

where word_emb($d$,300) retrieves 300-dimensional word embeddings for all words in a document, avg_pool executes average pooling, and the resulting vector is fed to a 4-layer MLP with Softsign. The model for CIFAR-10 was an *all convolutional net* [43]: (32*32*3)-[C(3*3,96)]*2-C(3*3,96,2)-[C(3*3,192)]*2-C(3*3,192,2)-C(3*3,192)-C(1*1,192)-C(1*1,10)-1000-1000-1, where the input is a 32*32 RGB image, C(3*3,96) means 96 channels of 3*3 convolutions followed by ReLU, [ · ]*2 means there are two such layers, C(3*3,96,2) means a similar layer but with stride 2, etc.; it is one of the best architectures for CIFAR-10. Batch normalization [44] was applied before hidden layers. Furthermore, the sigmoid loss $\ell_{\text{sig}}$ was used as the surrogate loss and an $\ell_2$-regularization was also added. The resulting objectives were minimized by Adam [20] on MNIST, epsilon and CIFAR-10, and by AdaGrad [31] on 20News; we fixed $\beta = 0$ and $\gamma = 1$ for simplicity.

The experimental results are reported in Figure 2, where means and standard deviations of training and test risks based on the same 10 random samplings are shown. We can see that uPU overfitted training data and nnPU fixed this problem. Additionally, given limited N data, nnPU outperformed PN on MNIST, epsilon and CIFAR-10 and was comparable to it on 20News. In summary, with the proposed non-negative risk estimator, we are able to use very flexible models given limited P data.

We further tried some cases where $\pi_{\text{p}}$ is misspecified, in order to simulate PU learning in the wild, where we must suffer from errors in estimating $\pi_{\text{p}}$. More specifically, we tested nnPU learning by replacing $\pi_{\text{p}}$ with $\pi'_{\text{p}} \in \{0.8\pi_{\text{p}}, 0.9\pi_{\text{p}}, \ldots, 1.2\pi_{\text{p}}\}$ and giving $\pi'_{\text{p}}$ to the learning method, so that it would regard $\pi'_{\text{p}}$ as $\pi_{\text{p}}$ during the entire training process. The experimental setup was exactly same as before except the replacement of $\pi_{\text{p}}$.

The experimental results are reported in Figure 3, where means of test risks of nnPU based on the same 10 random samplings are shown, and the best test risks are identified (horizontal lines are the best mean test risks and vertical lines are the epochs when they were achieved). We can see that on MNIST, the more misspecification was, the worse nnPU performed, while under-misspecification hurt more than over-misspecification; on epsilon, the cases where $\pi'_{\text{p}}$ equals to $\pi_{\text{p}}$, $1.1\pi_{\text{p}}$ and $1.2\pi_{\text{p}}$

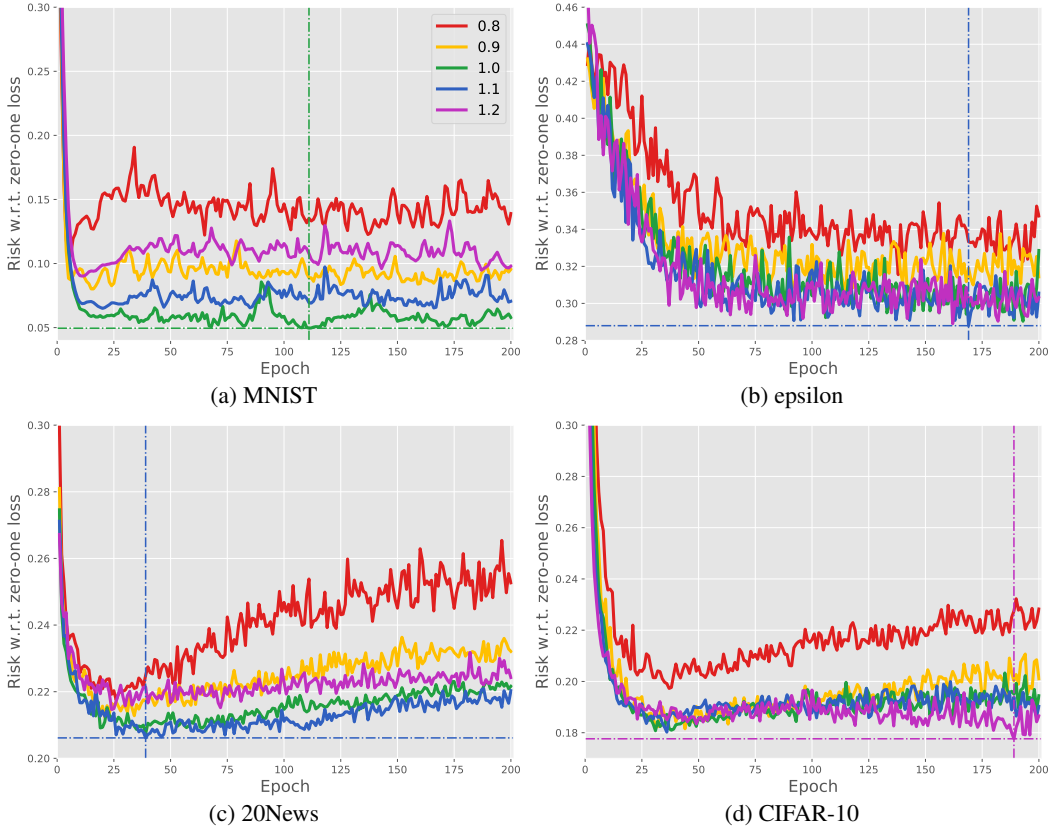

Figure 3: Experimental results given $\pi'_{\mathrm{p}} \in \{0.8\pi_{\mathrm{p}}, 0.9\pi_{\mathrm{p}}, \ldots, 1.2\pi_{\mathrm{p}}\}$.

were comparable, but the best was $\pi'_{\mathrm{p}} = 1.1\pi_{\mathrm{p}}$ rather than $\pi'_{\mathrm{p}} = \pi_{\mathrm{p}}$; on 20News, these three cases became different, such that $\pi'_{\mathrm{p}} = \pi_{\mathrm{p}}$ was superior to $\pi'_{\mathrm{p}} = 1.2\pi_{\mathrm{p}}$ but inferior to $\pi'_{\mathrm{p}} = 1.1\pi_{\mathrm{p}}$; at last on CIFAR-10, $\pi'_{\mathrm{p}} = \pi_{\mathrm{p}}$ and $\pi'_{\mathrm{p}} = 1.1\pi_{\mathrm{p}}$ were comparable again, and $\pi'_{\mathrm{p}} = 1.2\pi_{\mathrm{p}}$ was the winner.

In all the experiments, we have fixed $\beta = 0$, which may explain this phenomenon. Recall that uPU overfitted seriously on all the benchmark datasets, and note that the larger $\pi'_{\mathrm{p}}$ is, the more different nnPU is from uPU. Therefore, the replacement of $\pi_{\mathrm{p}}$ with some $\pi'_{\mathrm{p}} > \pi_{\mathrm{p}}$ introduces additional bias of $\widetilde{R}_{\mathrm{pu}}(g)$ in estimating $R(g)$, but it also pushes $\widetilde{R}_{\mathrm{pu}}(g)$ away from $\widehat{R}_{\mathrm{pu}}(g)$ and then pushes nnPU away from uPU. This may result in lower test risks given some $\pi'_{\mathrm{p}}$ slightly larger than $\pi_{\mathrm{p}}$ as shown in Figure 3. This is also why under-misspecified $\pi'_{\mathrm{p}}$ hurt more than over-misspecified $\pi'_{\mathrm{p}}$.

All the experiments were done with *Chainer* [45], and our implementation based on it is available at https://github.com/kiryor/nnPUlearning.

## 6 Conclusions

We proposed a non-negative risk estimator for PU learning that follows and improves on the state-of-the-art unbiased risk estimators. No matter how flexible the model is, it will not go negative as its unbiased counterparts. It is more robust against overfitting when being minimized, and training very flexible models such as deep neural networks given limited P data becomes possible. We also developed a large-scale PU learning algorithm. Extensive theoretical analyses were presented, and the usefulness of our non-negative PU learning was verified by intensive experiments. A promising future direction is extending the current work to semi-supervised learning along [46].

### Acknowledgments

GN and MS were supported by JST CREST JPMJCR1403 and GN was also partially supported by Microsoft Research Asia.

## Footnotes

[1]It implies the P and N class-conditional densities have disjoint support sets, and then any P and N data (as the test data) can be perfectly separated by a fixed classifier that is sufficiently flexible.

[2] $\mathcal{X}_\mathrm{p}$ is a set of independent data and so is $\mathcal{X}_\mathrm{u}$, but $\mathcal{X}_\mathrm{p} \cup \mathcal{X}_\mathrm{u}$ does not need to be such a set.

[3] The consistency here means for fixed $g$, $\widehat{R}_\mathrm{pn}(g) \to R(g)$ and $\widehat{R}_\mathrm{pu}(g) \to R(g)$ as $n_\mathrm{p}, n_\mathrm{n}, n_\mathrm{u} \to \infty$.

[4]Let $\sigma_1, \ldots, \sigma_n$ be $n$ Rademacher variables, the Rademacher complexity of $\mathcal{G}$ for $\mathcal{X}$ of size $n$ drawn from $q(x)$ is defined by $\mathfrak{R}_{n,q}(\mathcal{G}) = \mathbb{E}_{\mathcal{X}}\mathbb{E}_{\sigma_1,\ldots,\sigma_n}[\sup_{g \in \mathcal{G}} \frac{1}{n}\sum_{x_i \in \mathcal{X}} \sigma_i g(x_i)]$ [28]. For any fixed $\mathcal{G}$ and $q$, $\mathfrak{R}_{n,q}(\mathcal{G})$ still depends on $n$ and should decrease with $n$.

[5]Here, $\mathrm{MSE}(\cdot)$ is over repeated sampling of $(\mathcal{X}_\mathrm{p}, \mathcal{X}_\mathrm{u})$.

[6]This can be derived as $n_\mathrm{p}, n_\mathrm{u} \to \infty$ by applying the *central limit theorem* to the two differences and then *L'Hôpital's rule* to the ratio of *complementary error functions* [32].

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
