[Supplementary Material]

# A  Supplementary experimental results

Due to limited space, we considered the surrogate loss without the zero-one loss in Figure 1. Here, we include the zero-one loss and show the extended version of Figure 1 in Figure 4. In general, the curves of risks w.r.t. $\ell_{01}$ look quite similar to (but less smooth than) those w.r.t. $\ell_{\mathrm{sig}}$. Therefore, the curves of risks w.r.t. $\ell_{\mathrm{sig}}$ are more visually appealing as the illustrative experimental results.

(a) Linear model, risks w.r.t. $\ell_{\mathrm{sig}}$        (b) Linear model, risks w.r.t. $\ell_{01}$

(c) MLP, risks w.r.t. $\ell_{\mathrm{sig}}$        (d) MLP, risks w.r.t. $\ell_{01}$

Figure 4: The extended version of Figure 1.

# B  Proofs

In this appendix, we prove all the theoretical results in Section 4.

## B.1  Proof of Lemma 1

Let

$$p_{\mathrm{p}}(\mathcal{X}_{\mathrm{p}}) = p_{\mathrm{p}}(x_1^{\mathrm{p}}) \cdots p_{\mathrm{p}}(x_{n_{\mathrm{p}}}^{\mathrm{p}}), \quad p(\mathcal{X}_{\mathrm{u}}) = p(x_1^{\mathrm{u}}) \cdots p(x_{n_{\mathrm{u}}}^{\mathrm{u}})$$

be the probability density functions of $\mathcal{X}_{\mathrm{p}}$ and $\mathcal{X}_{\mathrm{u}}$. Then let $F_{\mathrm{p}}(\mathcal{X}_{\mathrm{p}})$ be the cumulative distribution function of $\mathcal{X}_{\mathrm{p}}$, $F_{\mathrm{u}}(\mathcal{X}_{\mathrm{u}})$ be that of $\mathcal{X}_{\mathrm{u}}$, and

$$F(\mathcal{X}_{\mathrm{p}}, \mathcal{X}_{\mathrm{u}}) = F_{\mathrm{p}}(\mathcal{X}_{\mathrm{p}}) \cdot F_{\mathrm{u}}(\mathcal{X}_{\mathrm{u}})$$

be the joint cumulative distribution function of $(\mathcal{X}_{\mathrm{p}}, \mathcal{X}_{\mathrm{u}})$. Given the above definitions, the measure of $\mathfrak{D}^-(g)$ is defined by

$$\Pr(\mathfrak{D}^-(g)) = \int_{(\mathcal{X}_{\mathrm{p}}, \mathcal{X}_{\mathrm{u}}) \in \mathfrak{D}^-(g)} \mathrm{d}F(\mathcal{X}_{\mathrm{p}}, \mathcal{X}_{\mathrm{u}}),$$

where Pr denotes the probability. Since $\widetilde{R}_{\mathrm{pu}}(g)$ is identical to $\widehat{R}_{\mathrm{pu}}(g)$ on $\mathfrak{D}^+(g)$ and different from $\widehat{R}_{\mathrm{pu}}(g)$ on $\mathfrak{D}^-(g)$, we have $\Pr(\mathfrak{D}^-(g)) = \Pr\{\widetilde{R}_{\mathrm{pu}}(g) \neq \widehat{R}_{\mathrm{pu}}(g)\}$. That is, the measure of $\mathfrak{D}^-(g)$ is non-zero if and only if $\widetilde{R}_{\mathrm{pu}}(g)$ differs from $\widehat{R}_{\mathrm{pu}}(g)$ with a non-zero probability.

Based on the facts that $\widehat{R}_{\mathrm{pu}}(g)$ is unbiased and $\widetilde{R}_{\mathrm{pu}}(g) - \widehat{R}_{\mathrm{pu}}(g) = 0$ on $\mathfrak{D}^+(g)$, we have

$$
\begin{aligned}
\mathbb{E}[\widetilde{R}_{\mathrm{pu}}(g)] - R(g) &= \mathbb{E}[\widetilde{R}_{\mathrm{pu}}(g) - \widehat{R}_{\mathrm{pu}}(g)] \\
&= \int_{(\mathcal{X}_{\mathrm{p}},\mathcal{X}_{\mathrm{u}})\in\mathfrak{D}^+(g)} \widetilde{R}_{\mathrm{pu}}(g) - \widehat{R}_{\mathrm{pu}}(g)\,\mathrm{d}F(\mathcal{X}_{\mathrm{p}},\mathcal{X}_{\mathrm{u}}) \\
&\quad + \int_{(\mathcal{X}_{\mathrm{p}},\mathcal{X}_{\mathrm{u}})\in\mathfrak{D}^-(g)} \widetilde{R}_{\mathrm{pu}}(g) - \widehat{R}_{\mathrm{pu}}(g)\,\mathrm{d}F(\mathcal{X}_{\mathrm{p}},\mathcal{X}_{\mathrm{u}}) \\
&= \int_{(\mathcal{X}_{\mathrm{p}},\mathcal{X}_{\mathrm{u}})\in\mathfrak{D}^-(g)} \widetilde{R}_{\mathrm{pu}}(g) - \widehat{R}_{\mathrm{pu}}(g)\,\mathrm{d}F(\mathcal{X}_{\mathrm{p}},\mathcal{X}_{\mathrm{u}}).
\end{aligned}
$$

As a result, $\mathbb{E}[\widetilde{R}_{\mathrm{pu}}(g)] - R(g) > 0$ if and only if $\int_{(\mathcal{X}_{\mathrm{p}},\mathcal{X}_{\mathrm{u}})\in\mathfrak{D}^-(g)} \mathrm{d}F(\mathcal{X}_{\mathrm{p}},\mathcal{X}_{\mathrm{u}}) > 0$ due to the fact $\widetilde{R}_{\mathrm{pu}}(g) - \widehat{R}_{\mathrm{pu}}(g) > 0$ on $\mathfrak{D}^-(g)$. That is, the bias of $\widetilde{R}_{\mathrm{pu}}(g)$ is positive if and only if the measure of $\mathfrak{D}^-(g)$ is non-zero.

We prove (7) by *the method of bounded differences*, for that

$$
\mathbb{E}[\widehat{R}_{\mathrm{u}}^-(g) - \pi_{\mathrm{p}}\widehat{R}_{\mathrm{p}}^-(g)] = R_{\mathrm{u}}^-(g) - \pi_{\mathrm{p}}R_{\mathrm{p}}^-(g) = R_{\mathrm{n}}^-(g) \geq \alpha.
$$

We have assumed that $0 \leq \ell(t,\pm 1) \leq C_\ell$, and thus the change of $\widehat{R}_{\mathrm{p}}^-(g)$ will be no more than $C_\ell/n_{\mathrm{p}}$ if some $x_i^{\mathrm{p}} \in \mathcal{X}_{\mathrm{p}}$ is replaced, or the change of $\widehat{R}_{\mathrm{u}}^-(g)$ will be no more than $C_\ell/n_{\mathrm{u}}$ if some $x_i^{\mathrm{u}} \in \mathcal{X}_{\mathrm{u}}$ is replaced. Subsequently, *McDiarmid's inequality* [47] implies

$$
\begin{aligned}
\Pr\{R_{\mathrm{n}}^-(g) - (\widehat{R}_{\mathrm{u}}^-(g) - \pi_{\mathrm{p}}\widehat{R}_{\mathrm{p}}^-(g)) \geq \alpha\} &\leq \exp\left(-\frac{2\alpha^2}{n_{\mathrm{p}}(C_\ell\pi_{\mathrm{p}}/n_{\mathrm{p}})^2 + n_{\mathrm{u}}(C_\ell/n_{\mathrm{u}})^2}\right) \\
&= \exp\left(-\frac{2\alpha^2/C_\ell^2}{\pi_{\mathrm{p}}^2/n_{\mathrm{p}} + 1/n_{\mathrm{u}}}\right).
\end{aligned}
$$

Taking into account that

$$
\begin{aligned}
\Pr(\mathfrak{D}^-(g)) &= \Pr\{\widehat{R}_{\mathrm{u}}^-(g) - \pi_{\mathrm{p}}\widehat{R}_{\mathrm{p}}^-(g) < 0\} \\
&\leq \Pr\{\widehat{R}_{\mathrm{u}}^-(g) - \pi_{\mathrm{p}}\widehat{R}_{\mathrm{p}}^-(g) \leq R_{\mathrm{n}}^-(g) - \alpha\} \\
&= \Pr\{R_{\mathrm{n}}^-(g) - (\widehat{R}_{\mathrm{u}}^-(g) - \pi_{\mathrm{p}}\widehat{R}_{\mathrm{p}}^-(g)) \geq \alpha\},
\end{aligned}
$$

we complete the proof. $\qquad\square$

### B.2  Proof of Theorem 2

It has been proven in Lemma 1 that

$$
\mathbb{E}[\widetilde{R}_{\mathrm{pu}}(g)] - R(g) = \int_{(\mathcal{X}_{\mathrm{p}},\mathcal{X}_{\mathrm{u}})\in\mathfrak{D}^-(g)} \widetilde{R}_{\mathrm{pu}}(g) - \widehat{R}_{\mathrm{pu}}(g)\,\mathrm{d}F(\mathcal{X}_{\mathrm{p}},\mathcal{X}_{\mathrm{u}}),
$$

and thus the exponential decay of the bias in (8) is obtained via

$$
\begin{aligned}
\mathbb{E}[\widetilde{R}_{\mathrm{pu}}(g)] - R(g) &\leq \sup_{(\mathcal{X}_{\mathrm{p}},\mathcal{X}_{\mathrm{u}})\in\mathfrak{D}^-(g)}(\widetilde{R}_{\mathrm{pu}}(g) - \widehat{R}_{\mathrm{pu}}(g)) \cdot \int_{(\mathcal{X}_{\mathrm{p}},\mathcal{X}_{\mathrm{u}})\in\mathfrak{D}^-(g)} \mathrm{d}F(\mathcal{X}_{\mathrm{p}},\mathcal{X}_{\mathrm{u}}) \\
&= \sup_{(\mathcal{X}_{\mathrm{p}},\mathcal{X}_{\mathrm{u}})\in\mathfrak{D}^-(g)}(\pi_{\mathrm{p}}\widehat{R}_{\mathrm{p}}^-(g) - \widehat{R}_{\mathrm{u}}^-(g)) \cdot \Pr(\mathfrak{D}^-(g)) \\
&\leq C_\ell\pi_{\mathrm{p}}\Delta_g.
\end{aligned}
$$

The deviation bound (9) is due to

$$
\begin{aligned}
|\widetilde{R}_{\mathrm{pu}}(g) - R(g)| &\leq |\widetilde{R}_{\mathrm{pu}}(g) - \mathbb{E}[\widetilde{R}_{\mathrm{pu}}(g)]| + |\mathbb{E}[\widetilde{R}_{\mathrm{pu}}(g)] - R(g)| \\
&\leq |\widetilde{R}_{\mathrm{pu}}(g) - \mathbb{E}[\widetilde{R}_{\mathrm{pu}}(g)]| + C_\ell\pi_{\mathrm{p}}\Delta_g.
\end{aligned}
$$

The change of $\widetilde{R}_{\mathrm{pu}}(g)$ will be no more than $2C_\ell/n_{\mathrm{p}}$ if some $x_i^{\mathrm{p}} \in \mathcal{X}_{\mathrm{p}}$ is replaced, or it will be no more than $C_\ell/n_{\mathrm{u}}$ if some $x_i^{\mathrm{u}} \in \mathcal{X}_{\mathrm{u}}$ is replaced, and McDiarmid's inequality gives us

$$\Pr\{|\widetilde{R}_{\mathrm{pu}}(g) - \mathbb{E}[\widetilde{R}_{\mathrm{pu}}(g)]| \geq \epsilon\} \leq 2\exp\left(-\frac{2\epsilon^2}{n_{\mathrm{p}}(2C_\ell\pi_{\mathrm{p}}/n_{\mathrm{p}})^2 + n_{\mathrm{u}}(C_\ell/n_{\mathrm{u}})^2}\right),$$

or equivalently, with probability at least $1 - \delta$,

$$|\widetilde{R}_{\mathrm{pu}}(g) - \mathbb{E}[\widetilde{R}_{\mathrm{pu}}(g)]| \leq \sqrt{\frac{\ln(2/\delta)C_\ell^2}{2}\left(\frac{4\pi_{\mathrm{p}}^2}{n_{\mathrm{p}}} + \frac{1}{n_{\mathrm{u}}}\right)}$$

$$\leq C_\delta\left(\frac{2\pi_{\mathrm{p}}}{\sqrt{n_{\mathrm{p}}}} + \frac{1}{\sqrt{n_{\mathrm{u}}}}\right)$$

$$= C_\delta \cdot \chi_{n_{\mathrm{p}},n_{\mathrm{u}}}.$$

On the other hand, the deviation bound (10) is due to

$$|\widetilde{R}_{\mathrm{pu}}(g) - R(g)| \leq |\widetilde{R}_{\mathrm{pu}}(g) - \widehat{R}_{\mathrm{pu}}(g)| + |\widehat{R}_{\mathrm{pu}}(g) - R(g)|,$$

where $|\widetilde{R}_{\mathrm{pu}}(g) - \widehat{R}_{\mathrm{pu}}(g)| > 0$ with probability at most $\Delta_g$, and $|\widehat{R}_{\mathrm{pu}}(g) - R(g)|$ shares the same concentration inequality with $|\widetilde{R}_{\mathrm{pu}}(g) - \mathbb{E}[\widetilde{R}_{\mathrm{pu}}(g)]|$. $\qquad\square$

## B.3 Proof of Theorem 3

For convenience, let $A = \pi_{\mathrm{p}}\widehat{R}_{\mathrm{p}}^+(g)$ and $B = \widehat{R}_{\mathrm{u}}^-(g) - \pi_{\mathrm{p}}\widehat{R}_{\mathrm{p}}^-(g)$, so that

$$R(g) = \mathbb{E}[A + B], \quad \widehat{R}_{\mathrm{pu}}(g) = A + B, \quad \widetilde{R}_{\mathrm{pu}}(g) = A + B_+,$$

where $B_+ = \max\{0, B\}$. Subsequently, let $R = R(g)$ for short, and then by definition,

$$\mathrm{MSE}(\widehat{R}_{\mathrm{pu}}(g)) = \mathbb{E}[(A + B - R)^2]$$
$$= \mathbb{E}[(A + B)^2] - 2R \cdot \mathbb{E}[A + B] + R^2,$$
$$\mathrm{MSE}(\widetilde{R}_{\mathrm{pu}}(g)) = \mathbb{E}[(A + B_+ - R(g))^2]$$
$$= \mathbb{E}[(A + B_+)^2] - 2R \cdot \mathbb{E}[A + B_+] + R^2.$$

Hence,

$$\mathrm{MSE}(\widehat{R}_{\mathrm{pu}}(g)) - \mathrm{MSE}(\widetilde{R}_{\mathrm{pu}}(g)) = \mathbb{E}[(A + B)^2] - \mathbb{E}[(A + B_+)^2]$$
$$- 2R \cdot (\mathbb{E}[A + B] - \mathbb{E}[A + B_+]).$$

The first part $\mathbb{E}[(A + B)^2] - \mathbb{E}[(A + B_+)^2]$ can be rewritten as

$$\mathbb{E}[(A + B)^2] - \mathbb{E}[(A + B_+)^2] = \mathbb{E}[2A(B - B_+) + B^2 - B_+^2]$$
$$= \int_{(\mathcal{X}_{\mathrm{p}},\mathcal{X}_{\mathrm{u}})\in\mathfrak{D}^+(g)} 2A(B - B) + B^2 - B^2\, \mathrm{d}F(\mathcal{X}_{\mathrm{p}}, \mathcal{X}_{\mathrm{u}})$$
$$+ \int_{(\mathcal{X}_{\mathrm{p}},\mathcal{X}_{\mathrm{u}})\in\mathfrak{D}^-(g)} 2A(B - 0) + B^2 - 0^2\, \mathrm{d}F(\mathcal{X}_{\mathrm{p}}, \mathcal{X}_{\mathrm{u}})$$
$$= \int_{(\mathcal{X}_{\mathrm{p}},\mathcal{X}_{\mathrm{u}})\in\mathfrak{D}^-(g)} 2AB + B^2\, \mathrm{d}F(\mathcal{X}_{\mathrm{p}}, \mathcal{X}_{\mathrm{u}}).$$

The second part $2R \cdot (\mathbb{E}[A + B] - \mathbb{E}[A + B_+])$ can be rewritten as

$$2R \cdot (\mathbb{E}[A + B] - \mathbb{E}[A + B_+]) = 2R \cdot \mathbb{E}[B - B_+]$$
$$= 2R \cdot \int_{(\mathcal{X}_{\mathrm{p}},\mathcal{X}_{\mathrm{u}})\in\mathfrak{D}^+(g)} B - B\, \mathrm{d}F(\mathcal{X}_{\mathrm{p}}, \mathcal{X}_{\mathrm{u}})$$
$$+ 2R \cdot \int_{(\mathcal{X}_{\mathrm{p}},\mathcal{X}_{\mathrm{u}})\in\mathfrak{D}^-(g)} B - 0\, \mathrm{d}F(\mathcal{X}_{\mathrm{p}}, \mathcal{X}_{\mathrm{u}})$$
$$= \int_{(\mathcal{X}_{\mathrm{p}},\mathcal{X}_{\mathrm{u}})\in\mathfrak{D}^-(g)} 2RB\, \mathrm{d}F(\mathcal{X}_{\mathrm{p}}, \mathcal{X}_{\mathrm{u}}).$$

As a consequence,

$$\mathrm{MSE}(\widehat{R}_{\mathrm{pu}}(g)) - \mathrm{MSE}(\widetilde{R}_{\mathrm{pu}}(g)) = \int_{(\mathcal{X}_{\mathrm{p}},\mathcal{X}_{\mathrm{u}})\in\mathfrak{D}^-(g)} (2A + B - 2R)B\,\mathrm{d}F(\mathcal{X}_{\mathrm{p}}, \mathcal{X}_{\mathrm{u}}),$$

which is exactly the left-hand side of (11) since $\widetilde{R}_{\mathrm{pu}}(g) = A$ on $\mathfrak{D}^-(g)$.

In order to prove the rest, it suffices to show that $A - R \le B$ on $\mathfrak{D}^-(g)$. By the assumption that $\ell$ satisfies (3),

$$A - R = A - \mathbb{E}[A] - \mathbb{E}[B]$$
$$= \pi_{\mathrm{p}}\widehat{R}_{\mathrm{p}}^+(g) - \pi_{\mathrm{p}}R_{\mathrm{p}}^+(g) - \mathbb{E}[B]$$
$$= \pi_{\mathrm{p}}R_{\mathrm{p}}^-(g) - \pi_{\mathrm{p}}\widehat{R}_{\mathrm{p}}^-(g) - \mathbb{E}[B].$$

Thus, with probability one,

$$A - R = \pi_{\mathrm{p}}R_{\mathrm{p}}^-(g) - \pi_{\mathrm{p}}\widehat{R}_{\mathrm{p}}^-(g) - \mathbb{E}[B] + (\widehat{R}_{\mathrm{u}}^-(g) - \widehat{R}_{\mathrm{u}}^-(g)) + (R_{\mathrm{u}}^-(g) - R_{\mathrm{u}}^-(g))$$
$$= (\widehat{R}_{\mathrm{u}}^-(g) - \pi_{\mathrm{p}}\widehat{R}_{\mathrm{p}}^-(g)) - (R_{\mathrm{u}}^-(g) - \pi_{\mathrm{p}}R_{\mathrm{p}}^-(g)) - \mathbb{E}[B] + (R_{\mathrm{u}}^-(g) - \widehat{R}_{\mathrm{u}}^-(g))$$
$$= B - 2\mathbb{E}[B] + (R_{\mathrm{u}}^-(g) - \widehat{R}_{\mathrm{u}}^-(g))$$
$$\le B,$$

where we used the assumptions that $\mathbb{E}[B] \ge \alpha$ and $R_{\mathrm{u}}^-(g) - \widehat{R}_{\mathrm{u}}^-(g) \le 2\alpha$ almost surely on $\mathfrak{D}^-(g)$. To sum up, we have established that

$$\int_{(\mathcal{X}_{\mathrm{p}},\mathcal{X}_{\mathrm{u}})\in\mathfrak{D}^-(g)} (2A + B - 2R)B\,\mathrm{d}F(\mathcal{X}_{\mathrm{p}}, \mathcal{X}_{\mathrm{u}}) \ge 3\int_{(\mathcal{X}_{\mathrm{p}},\mathcal{X}_{\mathrm{u}})\in\mathfrak{D}^-(g)} B^2\,\mathrm{d}F(\mathcal{X}_{\mathrm{p}}, \mathcal{X}_{\mathrm{u}}).$$

Due to the fact that $B^2 > 0$ on $\mathfrak{D}^-(g)$ and the assumption that $\mathrm{Pr}(\mathfrak{D}^-(g)) > 0$, we know Eq. (11) is valid. Finally, for any $0 \le \beta \le C_\ell\pi_{\mathrm{p}}$, it is clear that

$$\{(\mathcal{X}_{\mathrm{p}}, \mathcal{X}_{\mathrm{u}}) \mid B < -\beta\} \subseteq \{(\mathcal{X}_{\mathrm{p}}, \mathcal{X}_{\mathrm{u}}) \mid B < 0\} = \mathfrak{D}^-(g),$$

and $B < -\beta$ if and only if $\widetilde{R}_{\mathrm{pu}}(g) - \widehat{R}_{\mathrm{pu}}(g) > \beta$. These two facts imply that

$$\int_{(\mathcal{X}_{\mathrm{p}},\mathcal{X}_{\mathrm{u}})\in\mathfrak{D}^-(g)} B^2\,\mathrm{d}F(\mathcal{X}_{\mathrm{p}}, \mathcal{X}_{\mathrm{u}}) \ge \int_{(\mathcal{X}_{\mathrm{p}},\mathcal{X}_{\mathrm{u}})|B<-\beta} B^2\,\mathrm{d}F(\mathcal{X}_{\mathrm{p}}, \mathcal{X}_{\mathrm{u}})$$
$$\ge \beta^2\int_{(\mathcal{X}_{\mathrm{p}},\mathcal{X}_{\mathrm{u}})|B<-\beta} \mathrm{d}F(\mathcal{X}_{\mathrm{p}}, \mathcal{X}_{\mathrm{u}})$$
$$= \beta^2\mathrm{Pr}\{B < -\beta\}$$
$$= \beta^2\mathrm{Pr}\{\widetilde{R}_{\mathrm{pu}}(g) - \widehat{R}_{\mathrm{pu}}(g) > \beta\},$$

which proves (12) and the whole theorem. $\qquad\square$

## B.4 Proof of Lemma 5

**Preliminary** An alternative definition of the Rademacher complexity will be used in the proof:

$$\mathfrak{R}'_{n,q}(\mathcal{G}) = \mathbb{E}_{\mathcal{X}}\mathbb{E}_{\sigma_1,\ldots,\sigma_n}\left[\sup_{g\in\mathcal{G}}\left|\frac{1}{n}\sum_{x_i\in\mathcal{X}}\sigma_i g(x_i)\right|\right].$$

For the sake of comparison, the one we have used in the statements of theoretical results is

$$\mathfrak{R}_{n,q}(\mathcal{G}) = \mathbb{E}_{\mathcal{X}}\mathbb{E}_{\sigma_1,\ldots,\sigma_n}\left[\sup_{g\in\mathcal{G}}\frac{1}{n}\sum_{x_i\in\mathcal{X}}\sigma_i g(x_i)\right].$$

This alternative version comes from [35, 36] of which authors are the pioneers of error bounds based on the Rademacher complexity. Without any composition, $\mathfrak{R}'_{n,q}(\mathcal{G}) \ge \mathfrak{R}_{n,q}(\mathcal{G})$ for arbitrary $\mathcal{G}$ and $\mathfrak{R}'_{n,q}(\mathcal{G}) = \mathfrak{R}_{n,q}(\mathcal{G})$ if $\mathcal{G}$ is closed under negation. However, with a composition

$$\ell \circ \mathcal{G} = \{\ell \circ g \mid g \in \mathcal{G}\}$$

where the loss $\ell$ is non-negative, the Rademacher complexity of the *composite function class* would generally not satisfy $\mathfrak{R}'_{n,q}(\ell \circ \mathcal{G}) = \mathfrak{R}_{n,q}(\ell \circ \mathcal{G})$ since $\ell \circ \mathcal{G}$ is generally not closed under negation. Furthermore, a vital disagreement arises when considering the contraction principle or property: if $\psi : \mathbb{R} \to \mathbb{R}$ is a Lipschitz continuous function with a Lipschitz constant $L_\psi$ and satisfies $\psi(0) = 0$, we have

$$\mathfrak{R}_{n,q}(\psi \circ \mathcal{G}) \le L_\psi \mathfrak{R}_{n,q}(\mathcal{G}),$$
$$\mathfrak{R}'_{n,q}(\psi \circ \mathcal{G}) \le 2L_\psi \mathfrak{R}'_{n,q}(\mathcal{G}),$$

according to *Talagrand's contraction lemma* [48] and its extension [28, 49]. Here, for $\mathfrak{R}_{n,q}(\psi \circ \mathcal{G})$ we can use Lemma 4.2 in [28] or Lemma 26.9 in [49] where $\psi(0) = 0$ is safely dropped, while for $\mathfrak{R}'_{n,q}(\psi \circ \mathcal{G})$ we have to use the original Theorem 4.12 in [48] where $\psi(0) = 0$ is required. In fact, the name of the lemma is after that $\psi$ is a contraction if $\psi(0) = 0$ and $L_\psi = 1$.

**Proof**   Firstly, we deal with the bias of $\widetilde{R}_{\mathrm{pu}}(g)$:

$$\sup_{g \in \mathcal{G}} |\widetilde{R}_{\mathrm{pu}}(g) - R(g)| \le \sup_{g \in \mathcal{G}} |\widetilde{R}_{\mathrm{pu}}(g) - \mathbb{E}[\widetilde{R}_{\mathrm{pu}}(g)]| + \sup_{g \in \mathcal{G}} |\mathbb{E}[\widetilde{R}_{\mathrm{pu}}(g)] - R(g)|$$
$$\le \sup_{g \in \mathcal{G}} |\widetilde{R}_{\mathrm{pu}}(g) - \mathbb{E}[\widetilde{R}_{\mathrm{pu}}(g)]| + C_\ell \pi_{\mathrm{p}} \Delta, \tag{16}$$

where we followed the assumption that $\inf_{g \in \mathcal{G}} R_{\mathrm{n}}^-(g) \ge \alpha > 0$ and Theorem 2.

Secondly, we apply McDiarmid's inequality to the uniform deviation $\sup_{g \in \mathcal{G}} |\widetilde{R}_{\mathrm{pu}}(g) - \mathbb{E}[\widetilde{R}_{\mathrm{pu}}(g)]|$ to get that with probability at least $1 - \delta$,

$$\sup_{g \in \mathcal{G}} |\widetilde{R}_{\mathrm{pu}}(g) - \mathbb{E}[\widetilde{R}_{\mathrm{pu}}(g)]| - \mathbb{E}[\sup_{g \in \mathcal{G}} |\widetilde{R}_{\mathrm{pu}}(g) - \mathbb{E}[\widetilde{R}_{\mathrm{pu}}(g)]|] \le C'_\delta \cdot \chi_{n_{\mathrm{p}}, n_{\mathrm{u}}}. \tag{17}$$

Notice that this concentration inequality is single-sided even though the uniform deviation itself is double-sided, which is different from the non-uniform deviation in Theorem 2.

Thirdly, we make *symmetrization* [50]. Suppose that $(\mathcal{X}'_{\mathrm{p}}, \mathcal{X}'_{\mathrm{u}})$ is a *ghost sample*, then

$$\mathbb{E}[\sup_{g \in \mathcal{G}} |\widetilde{R}_{\mathrm{pu}}(g) - \mathbb{E}[\widetilde{R}_{\mathrm{pu}}(g)]|] = \mathbb{E}_{(\mathcal{X}_{\mathrm{p}}, \mathcal{X}_{\mathrm{u}})}[\sup_{g \in \mathcal{G}} |\widetilde{R}_{\mathrm{pu}}(g) - \mathbb{E}_{(\mathcal{X}'_{\mathrm{p}}, \mathcal{X}'_{\mathrm{u}})}[\widetilde{R}_{\mathrm{pu}}(g)]|]$$
$$\le \mathbb{E}_{(\mathcal{X}_{\mathrm{p}}, \mathcal{X}_{\mathrm{u}}), (\mathcal{X}'_{\mathrm{p}}, \mathcal{X}'_{\mathrm{u}})}[\sup_{g \in \mathcal{G}} |\widetilde{R}_{\mathrm{pu}}(g; \mathcal{X}_{\mathrm{p}}, \mathcal{X}_{\mathrm{u}}) - \widetilde{R}_{\mathrm{pu}}(g; \mathcal{X}'_{\mathrm{p}}, \mathcal{X}'_{\mathrm{u}})|],$$

where we applied *Jensen's inequality* twice since the absolute value and the supremum are convex. By decomposing the difference $|\widetilde{R}_{\mathrm{pu}}(g; \mathcal{X}_{\mathrm{p}}, \mathcal{X}_{\mathrm{u}}) - \widetilde{R}_{\mathrm{pu}}(g; \mathcal{X}'_{\mathrm{p}}, \mathcal{X}'_{\mathrm{u}})|$, we can know that

$$|\widetilde{R}_{\mathrm{pu}}(g; \mathcal{X}_{\mathrm{p}}, \mathcal{X}_{\mathrm{u}}) - \widetilde{R}_{\mathrm{pu}}(g; \mathcal{X}'_{\mathrm{p}}, \mathcal{X}'_{\mathrm{u}})|$$
$$= |\pi_{\mathrm{p}} \widehat{R}_{\mathrm{p}}^+(g; \mathcal{X}_{\mathrm{p}}) - \pi_{\mathrm{p}} \widehat{R}_{\mathrm{p}}^+(g; \mathcal{X}'_{\mathrm{p}})$$
$$+ \max\{0, \widehat{R}_{\mathrm{u}}^-(g; \mathcal{X}_{\mathrm{u}}) - \pi_{\mathrm{p}} \widehat{R}_{\mathrm{p}}^-(g; \mathcal{X}_{\mathrm{p}})\} - \max\{0, \widehat{R}_{\mathrm{u}}^-(g; \mathcal{X}'_{\mathrm{u}}) - \pi_{\mathrm{p}} \widehat{R}_{\mathrm{p}}^-(g; \mathcal{X}'_{\mathrm{p}})\}|$$
$$\le \pi_{\mathrm{p}} |\widehat{R}_{\mathrm{p}}^+(g; \mathcal{X}_{\mathrm{p}}) - \widehat{R}_{\mathrm{p}}^+(g; \mathcal{X}'_{\mathrm{p}})| + \pi_{\mathrm{p}} |\widehat{R}_{\mathrm{p}}^-(g; \mathcal{X}_{\mathrm{p}}) - \widehat{R}_{\mathrm{p}}^-(g; \mathcal{X}'_{\mathrm{p}})| + |\widehat{R}_{\mathrm{u}}^-(g; \mathcal{X}_{\mathrm{u}}) - \widehat{R}_{\mathrm{u}}^-(g; \mathcal{X}'_{\mathrm{u}})|$$

where we employed $|\max\{0, z\} - \max\{0, z'\}| \le |z - z'|$. This decomposition results in

$$\mathbb{E}[\sup_{g \in \mathcal{G}} |\widetilde{R}_{\mathrm{pu}}(g) - \mathbb{E}[\widetilde{R}_{\mathrm{pu}}(g)]|] \le \pi_{\mathrm{p}} \mathbb{E}_{\mathcal{X}_{\mathrm{p}}, \mathcal{X}'_{\mathrm{p}}}[\sup_{g \in \mathcal{G}} |\widehat{R}_{\mathrm{p}}^+(g; \mathcal{X}_{\mathrm{p}}) - \widehat{R}_{\mathrm{p}}^+(g; \mathcal{X}'_{\mathrm{p}})|]$$
$$+ \pi_{\mathrm{p}} \mathbb{E}_{\mathcal{X}_{\mathrm{p}}, \mathcal{X}'_{\mathrm{p}}}[\sup_{g \in \mathcal{G}} |\widehat{R}_{\mathrm{p}}^-(g; \mathcal{X}_{\mathrm{p}}) - \widehat{R}_{\mathrm{p}}^-(g; \mathcal{X}'_{\mathrm{p}})|]$$
$$+ \mathbb{E}_{\mathcal{X}_{\mathrm{u}}, \mathcal{X}'_{\mathrm{u}}}[\sup_{g \in \mathcal{G}} |\widehat{R}_{\mathrm{u}}^-(g; \mathcal{X}_{\mathrm{u}}) - \widehat{R}_{\mathrm{u}}^-(g; \mathcal{X}'_{\mathrm{u}})|]. \tag{18}$$

Fourthly, we relax those expectations in (18) to Rademacher complexities. The original $\ell$ may miss the origin, i.e., $\ell(0, y) \ne 0$, with which we need to cope. Let

$$\tilde{\ell}(t, y) = \ell(t, y) - \ell(0, y)$$

be a *shifted loss* so that $\tilde{\ell}(0, y) = 0$. Note that for all $t, t' \in \mathbb{R}$ and $y = \pm 1$,

$$\ell(t, y) - \ell(t', y) = \tilde{\ell}(t, y) - \tilde{\ell}(t', y).$$

Hence,

$$
\begin{aligned}
\widehat{R}_{\mathrm{p}}^+(g; \mathcal{X}_{\mathrm{p}}) - \widehat{R}_{\mathrm{p}}^+(g; \mathcal{X}_{\mathrm{p}}') &= (1/n_{\mathrm{p}}) \sum_{x_i \in \mathcal{X}_{\mathrm{p}}} \ell(g(x_i), +1) - (1/n_{\mathrm{p}}) \sum_{x_i' \in \mathcal{X}_{\mathrm{p}}'} \ell(g(x_i'), +1) \\
&= (1/n_{\mathrm{p}}) \sum_{i=1}^{n_{\mathrm{p}}} (\ell(g(x_i), +1) - \ell(g(x_i'), +1)) \\
&= (1/n_{\mathrm{p}}) \sum_{i=1}^{n_{\mathrm{p}}} (\tilde{\ell}(g(x_i), +1) - \tilde{\ell}(g(x_i'), +1)).
\end{aligned}
$$

This is already a standard form where we can attach Rademacher variables to every $\tilde{\ell}(g(x_i), +1) - \tilde{\ell}(g(x_i'), +1)$, and it is a routine work to show that

$$
\mathbb{E}_{\mathcal{X}_{\mathrm{p}}, \mathcal{X}_{\mathrm{p}}'} [\sup_{g \in \mathcal{G}} |\widehat{R}_{\mathrm{p}}^+(g; \mathcal{X}_{\mathrm{p}}) - \widehat{R}_{\mathrm{p}}^+(g; \mathcal{X}_{\mathrm{p}}')|] \leq 2\mathfrak{R}_{n_{\mathrm{p}}, p_{\mathrm{p}}}(\tilde{\ell}(\cdot, +1) \circ \mathcal{G}).
$$

The other two expectations can be handled analogously. As a result, (18) can be reduced to

$$
\begin{aligned}
\mathbb{E}[\sup_{g \in \mathcal{G}} |\widetilde{R}_{\mathrm{pu}}(g) - \mathbb{E}[\widetilde{R}_{\mathrm{pu}}(g)]|] \leq{}& 2\pi_{\mathrm{p}} \mathfrak{R}'_{n_{\mathrm{p}}, p_{\mathrm{p}}}(\tilde{\ell}(\cdot, +1) \circ \mathcal{G}) \\
&+ 2\pi_{\mathrm{p}} \mathfrak{R}'_{n_{\mathrm{p}}, p_{\mathrm{p}}}(\tilde{\ell}(\cdot, -1) \circ \mathcal{G}) + 2\mathfrak{R}'_{n_{\mathrm{u}}, p}(\tilde{\ell}(\cdot, -1) \circ \mathcal{G}). \quad (19)
\end{aligned}
$$

Finally, we transform the Rademacher complexities of composite function classes in (19) to those of the original function class. It is obvious that $\tilde{\ell}$ shares the same Lipschitz constant $L_\ell$ with $\ell$, and consequently

$$
\begin{aligned}
\mathfrak{R}'_{n_{\mathrm{p}}, p_{\mathrm{p}}}(\tilde{\ell}(\cdot, +1) \circ \mathcal{G}) &\leq 2L_\ell \mathfrak{R}'_{n_{\mathrm{p}}, p_{\mathrm{p}}}(\mathcal{G}) = 2L_\ell \mathfrak{R}_{n_{\mathrm{p}}, p_{\mathrm{p}}}(\mathcal{G}) \\
\mathfrak{R}'_{n_{\mathrm{p}}, p_{\mathrm{p}}}(\tilde{\ell}(\cdot, -1) \circ \mathcal{G}) &\leq 2L_\ell \mathfrak{R}'_{n_{\mathrm{p}}, p_{\mathrm{p}}}(\mathcal{G}) = 2L_\ell \mathfrak{R}_{n_{\mathrm{p}}, p_{\mathrm{p}}}(\mathcal{G}) \\
\mathfrak{R}'_{n_{\mathrm{u}}, p}(\tilde{\ell}(\cdot, -1) \circ \mathcal{G}) &\leq 2L_\ell \mathfrak{R}'_{n_{\mathrm{u}}, p}(\mathcal{G}) = 2L_\ell \mathfrak{R}_{n_{\mathrm{u}}, p}(\mathcal{G}),
\end{aligned} \quad (20)
$$

where we used Talagrand's contraction lemma and the assumption that $\mathcal{G}$ is closed under negation. Combining (16), (17), (19) and (20) finishes the proof of the uniform deviation bound (15). $\qquad\square$

## B.5   Proof of Theorem 4

Based on Lemma 5, the estimation error bound (13) is proven through

$$
\begin{aligned}
R(\widetilde{g}_{\mathrm{pu}}) - R(g^*) &= \left( \widetilde{R}_{\mathrm{pu}}(\widetilde{g}_{\mathrm{pu}}) - \widetilde{R}_{\mathrm{pu}}(g^*) \right) + \left( R(\widetilde{g}_{\mathrm{pu}}) - \widetilde{R}_{\mathrm{pu}}(\widetilde{g}_{\mathrm{pu}}) \right) + \left( \widetilde{R}_{\mathrm{pu}}(g^*) - R(g^*) \right) \\
&\leq 0 + 2 \sup_{g \in \mathcal{G}} |\widetilde{R}_{\mathrm{pu}}(g) - R(g)| \\
&\leq 16 L_\ell \pi_{\mathrm{p}} \mathfrak{R}_{n_{\mathrm{p}}, p_{\mathrm{p}}}(\mathcal{G}) + 8 L_\ell \mathfrak{R}_{n_{\mathrm{u}}, p}(\mathcal{G}) + 2C_\delta' \cdot \chi_{n_{\mathrm{p}}, n_{\mathrm{u}}} + 2C_\ell \pi_{\mathrm{p}} \Delta,
\end{aligned}
$$

where $\widetilde{R}_{\mathrm{pu}}(\widetilde{g}_{\mathrm{pu}}) \leq \widetilde{R}_{\mathrm{pu}}(g^*)$ by the definition of $\widetilde{g}_{\mathrm{pu}}$. $\qquad\square$