[Reviews · NeurIPS 2017]

Reviewer 1



This paper presents a new risk estimator aiming to solve the problem that the unbiased PU learning risk estimator is unbounded from below such that the model can overfitt easily especially when the model is complex. Theoretical analysis and empricial studies are conducted and results are reported. In PU learning, unbiased PU risk estimator is not bounded below and might lead to serious overfitting . This paper tackles this problem by slightly adjust the original estimators by setting the a cut off at zero. Such modification is simple, but effective. The authors studies the theoretical properties of the revised estimators and showed that this estimator would be effective especially when the number of unlabeled data is abundant. The experiments on several datasets shows that performance of PU learning improves after using the new risk estimator. Minor issues: the presentation of this paper can be further improved.

Reviewer 2



Summary The paper builds on the literature of Positive and Unlabeled (PU) learning and formulates a new risk objective that is negatively bounded. The need is mainly motivated by the fact that the state of the art risk formulation for PU can be unbounded from below. This is a serious issue when dealing with models with high capacity (such as deep neural networks), because the model can overfit. A new lower-bounded formulation is provided. Despite not being unbiased, the new risk is consistent and its bias decreases exponentially with the sample size. Generalization bounds are also proven. Experiments confirm that state of the art cannot be used with high-capacity models, while the new risk is robust to overfitting and can be applied for classification with simple deep neural networks. Comments The paper is very well written, easy to follow and tells a coherent story. The theoretical results are flawless from my point of view; I have checked the proofs in the supplementary material. The theory is also supported by many comments helping intuition of the arguments behind the proofs. This is very good. I am only confused by the meaning of those two sentences in Section 4: - Theorem 3 is not a necessary condition even for meeting (3): what is the condition the authors refer to? - In the proof, the reduction in MSE was expressed as a Lebesgue-Stieltjes integral [29]: why is this relevant? The authors successfully show that optimisation of this new risk formulation may be effective in practice, on deep neural networks, with only a few (positively) labeled data points. From this, it could help to see what is the point of view of the authors in comparing PU with current trends in deep learning, such as few-shots learning. The dataset ε is never mentioned before talking about the model trained on it; I find that confusing for the reader. Experimental results well address the main point of the paper (i.e. loweboundness/overfitting), but could be expanded. Fig. 1 shows two graphs with a very similar information: PU overfits when training a deep net. A curious reader who does not know about [16] could wonder whether the previous approach actually works at all. Why not showing the same graph (for one among the 2 losses) on a scenario where [16] is effective? For example, given the thesis of the paper, [16] should work fine if we train a linear model on MNIST, right? Additionally, it would be interesting to see what is the effect of estimating the class-prior on performance. Experiments in Fig. 2 are designed to show that NNPU can be even better than PN. But what happens if the class priors are estimated by a state of the art method? Since the paper is mostly focus on the theory side, I think this lack is not a major issue for acceptance. Although it will be for any realistic implementation of the proposed method. Minor 53: Menon et al. ICML 2015 is another valid method for estimating class priors other than [22, 23, 24] 56: Curiosity on notation: why not denoting R_n(g) as R_n^-(g) ? That would be symmetric with R_n^+ (same for the empirical risks) 237: performance of [35] on CIFAR10 has been surpassed, see https://www.eff.org/ai/metrics or http://rodrigob.github.io/are_we_there_yet/build/classification_datasets_results.html#43494641522d3130 I think it might be said that [35] is "among the best performing architectures for classifying on CIFAR10"

Reviewer 3



This paper considers PU learning, and particularly, unbiased risk estimators for PU learning. The authors convincingly show that because the unbiased risk estimators can be negative, that this leads to poor test errors. Figure 1 is very illustrative, and demonstrates that the test error tends to diverge when the training error drops below zero. This fact is surprising when considering lemma 1, which shows that for a fixed classifier, the unbiased risk estimator is positive. Apparently, for the rare classifiers when this is not true the test error is very poor. The correction is very simple in that they do not allow a weighted difference between the unlabeled risk and the false positive risk to exceed 0. The authors propose a stochastic gradient procedure based on this non-negative risk estimator. Lemma 1 implies that the bias and mean square error of the NNRE is controlled. Theorem 4 explores the estimation error based on Rademacher complexity of the classifiers. The experiments conclusively show that this methodology improves pre-existing PU learning methods. I have gone over the proofs in the supplement and cannot find faults.